# Information-to-work conversion by Maxwell's demon in a superconducting circuit quantum electrodynamical system

Y. Masuyama [1], K. Funo[2], Y. Murashita[3], A. Noguchi[1], S. Kono[1], Y. Tabuchi [1], R. Yamazaki[1], M. Ueda[3,4] & Y. Nakamura [1,4]

Information thermodynamics bridges information theory and statistical physics by connecting information content and entropy production through measurement and feedback control. Maxwell's demon is a hypothetical character that uses information about a system to reduce its entropy. Here we realize a Maxwell's demon acting on a superconducting quantum circuit. We implement quantum non-demolition projective measurement and feedback operation of a qubit and verify the generalized integral fluctuation theorem. We also evaluate the conversion efficiency from information gain to work in the feedback protocol. Our experiment constitutes a step toward experimental studies of quantum information thermodynamics in artificially made quantum machines.

[1] Research Center for Advanced Science and Technology (RCAST), The University of Tokyo, 4-6-1 Komaba, Meguro-ku, Tokyo 153-8904, Japan. [2] School of Physics, Peking University, Beijing 100871, China. [3] Department of Physics, The University of Tokyo, 7-3-1 Hongo, Bunkyo-ku, Tokyo 113-0033, Japan. [4] Center for Emergent Matter Science (CEMS), RIKEN, 2-1 Hirosawa, Wako, Saitama 351-0198, Japan. Correspondence and requests for materials should be addressed to Y.N. (email: yasunobu@ap.t.u-tokyo.ac.jp)

The gedanken experiment of Maxwell's demon has led to the studies concerning the foundations of thermodynamics and statistical mechanics[1]. The demon measures fluctuations of a system's observable and converts the information gain into work via feedback control[2]. Recent developments in information thermodynamics have elucidated the relationship between the acquired information and the entropy production and generalized the second law of thermodynamics and the fluctuation theorems[3–6]. Here we extend the scope to a system subject to quantum fluctuations by exploiting techniques in superconducting circuit quantum electrodynamics[7]. We implement Maxwell's demon equipped with coherent control and quantum non-demolition (QND) projective measurements on a superconducting qubit, thereby verifying the generalized integral fluctuation theorems[8,9] and the information-to-work conversion. This demonstrates the potential of superconducting circuits as a versatile platform for investigating quantum information thermodynamics under feedback control, which may find applications to quantum error correction[10] for computation[11] and metrology[12].

The fluctuation theorem is valid in systems far from equilibrium and can be regarded as a generalization of the second law of thermodynamics and the fluctuation–dissipation theorem[13,14]. In particular, the generalized integral fluctuation theorem, which incorporates the information content on equal footing with the entropy production, bridges information theory and statistical mechanics[15], and has been extended to quantum systems[9,16]. Experimentally, Maxwell's demons were implemented in classical systems using colloidal particles[4], a single electron box[5], and a photodetector[6]. More recently, the integral quantum fluctuation theorem in the absence of feedback control was tested with a trapped ion[17]. Maxwell's demon and the generalized second law in a quantum system were studied in spin ensembles with nuclear magnetic resonance spectroscopy[18]. However, experimental demonstrations of the fluctuation theorems that directly address the statistics of single quantum trajectories under feedback control are still elusive. Toward this goal, recent progress in superconducting quantum circuits offers a QND projective measurement of a qubit[7] and a sufficiently long coherence time[19], which altogether enable high-fidelity feedback operations. For example, stabilization of Rabi oscillations using coherent feedback[20,21], fast initialization of a qubit[22], and deterministic generation of an entangled state between two qubits[23] have been achieved.

Here we verify the generalized integral fluctuation theorem under feedback control by using a superconducting transmon qubit as a quantum system and taking statistics over repeated single-shot measurements on individual quantum trajectories. It is noteworthy that Naghiloo et al.[24] recently reported a related experiment with continuous weak measurement and feedback. We first investigate the role of absolute irreversibility associated with a projective measurement and feedback control[8], and then study the effect of imperfect projection.

## Results

**Absolute irreversibility**. The fluctuation theorem is formulated by considering a pair of processes, the original forward process and its time-reversed reference process, both of which are assumed to start with the canonical distribution at the same temperature $T$. Figure 1a illustrates an example of such processes. If we consider an ideal projective measurement and ignore relaxation of the qubit, the fluctuation theorem reads[8] (see also Supplementary Note 1)

$$\langle e^{-\sigma - I_{\mathrm{Sh}}} \rangle_{\mathrm{PM}} = 1 - \lambda_{\mathrm{fb}}, \tag{1}$$

where $I_{\mathrm{Sh}}$ is the stochastic Shannon entropy the demon acquires in the projective measurement, $\sigma = -\beta(W + \Delta F)$ is the entropy production, $\beta$ is the inverse temperature $1/(k_{\mathrm{B}}T)$ of the initial state of the qubit, $W$ is the work extracted from the qubit via the feedback operation $\hat{U}$, and $\Delta F$ is the change in the equilibrium free energy of the system. The angle brackets $\langle \cdot \rangle_{\mathrm{PM}}$ indicate the statistical average obtained with a protocol using a projective measurement for the feedback control. Below we focus on the case with $\Delta F = 0$, i.e., on the process with the same system Hamiltonian at the beginning and the end, for simplicity of discussions.

The constant $\lambda_{\mathrm{fb}}$ on the right-hand side of Eq. (1) gives the total probability of those events in the time-reversed process, whose counterparts in the original process do not exist. Such events, which we call absolutely irreversible events, involve a formal divergence of the entropy production and should therefore be treated separately[8] (see also Supplementary Note 1). Here, the absolute irreversibility is caused by the combination of the projective measurement that restricts possible forward events and the non-ideal property of the feedback operation that makes the backward events random. For example, in the process shown in Fig. 1a, the projective measurement and the feedback operation,

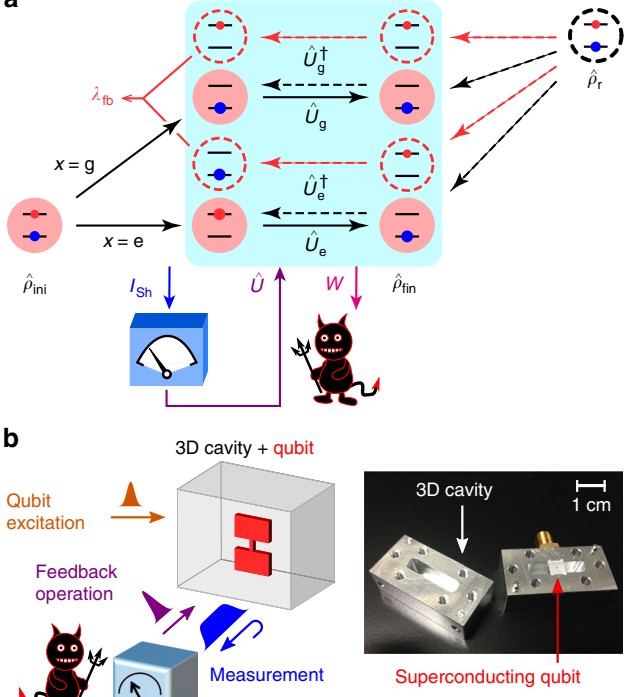

**Fig. 1** Maxwell's demon and absolute irreversibility. **a** Concept of the experiment. The system initially prepared in a canonical distribution $\hat{\rho}_{\mathrm{ini}}$ evolves in time. A projective measurement [with outcome $x(=\mathrm{g}$ or $\mathrm{e})$] by the demon projects the system onto a quantum state. The demon gains the stochastic Shannon entropy $I_{\mathrm{Sh}}$ and converts it into work $W$ via a feedback operation $\hat{U}$ ($=\hat{U}_{\mathrm{g}}$ or $\hat{U}_{\mathrm{e}}$). The forward process ends up in the final distribution $\hat{\rho}_{\mathrm{fin}}$. The time-reversed reference process starts from a reference state $\hat{\rho}_{\mathrm{r}}$, which we choose to be equal to $\hat{\rho}_{\mathrm{ini}}$. The absolute irreversibility is quantified with $\lambda_{\mathrm{fb}}$, the probability of those events in the time-reversed process whose counterparts in the original process do not exist (red dashed arrows). **b** Schematic of the feedback-controlled system in the experiment. The right panel shows a photograph of the qubit-resonator coupled system. The cavity resonator is disassembled to show its internal structure

$\hat{U}_g$ or $\hat{U}_e$, always bring the system to the ground state. Therefore, the evolution of the excited state in the reverse process via the operation $\hat{U}_g^\dagger$ or $\hat{U}_e^\dagger$ does not have a counterpart in the forward process. The probability $\lambda_{fb}$ of such events in the present protocol is given by $e^{-\beta\hbar\omega_q}/(1 + e^{-\beta\hbar\omega_q})$, i.e., the excited state occupation probability in $\hat{\rho}_r$.

The absolute irreversibility makes a significant contribution to the generalized second law of thermodynamics including the effect of the feedback control. For achieving the ultimate bound on the extracted work $\langle W \rangle_{PM} = k_B T \langle I_{Sh} \rangle_{PM}$, the final state distribution $\hat{\rho}_{fin}$ of the system has to be the same as $\hat{\rho}_{ini}$[3, 8]. However, the projective measurement together with the unoptimized feedback operation prevents it and limits the amount of the extractable work (see Eq. (4) below).

In our experiment, a superconducting transmon qubit (i.e., the system) is placed at the center of an aluminum-made superconducting cavity resonator (Fig. 1b). The qubit state is controlled with a resonant microwave pulse, which induces Rabi rotation. Owing to the interaction between the qubit and the detuned resonator, the resonance frequency of the resonator varies depending on the qubit state. We utilize this property for the QND readout of the qubit; the ground and excited states are distinguished in the phase shift of a readout microwave pulse reflected by the resonator[7].

**Protocol with projective measurements.** Figure 2a shows the sequence of the experiment corresponding to Fig. 1a. The qubit is initialized with a projective measurement and postselection, followed by a resonant pulse excitation, which prepares as an input a superposition state $\alpha_g|g\rangle + \alpha_e|e\rangle$ $\left(|\alpha_g|^2 + |\alpha_e|^2 = 1\right)$ of the ground ($|g\rangle$) and excited ($|e\rangle$) states of the qubit. As the qubit is subject to the subsequent projective measurement, the coherence in the input state does not have any essential role here, and the coefficients of the superposition define the effective temperature of the system $T = (\hbar\omega_q/k_B)/\ln\left(|\alpha_g|^2/|\alpha_e|^2\right)$ after the projection, where $\hbar\omega_q$ is the qubit excitation energy.

We evaluate the work $W(x, z) = E(x) - E(z)$ extracted from the system by employing the two-point measurement protocol (TPM), in which QND projective measurements on the energy eigenbasis with outcomes $x(= g \text{ or } e)$ and $z(= g \text{ or } e)$ are applied respectively to the initial and final states of the system[14]. Here $E$(g) and $E$(e) denotes the energies of the qubit in the ground and excited states, respectively. Depending on the measurement outcome $x$ for the feedback control, the feedback operation does or does not flip the state of the qubit with a $\pi$-pulse. A positive amount of the work ($W > 0$) implies that the energy is extracted from the system via the stimulated emission of a single photon induced by a $\pi$-pulse, which flips the qubit state. The probability $p(x)$ of the state $x$ being found gives $I_{Sh}(x) = -\ln p(x)$.

In Fig. 2b we compare the experimentally obtained statistical average $\left\langle e^{\beta W - I_{Sh}} \right\rangle_{PM} = \sum_{x,z} p(x, z) e^{\beta W(x,z) - I_{Sh}(x)}$ (blue circles)

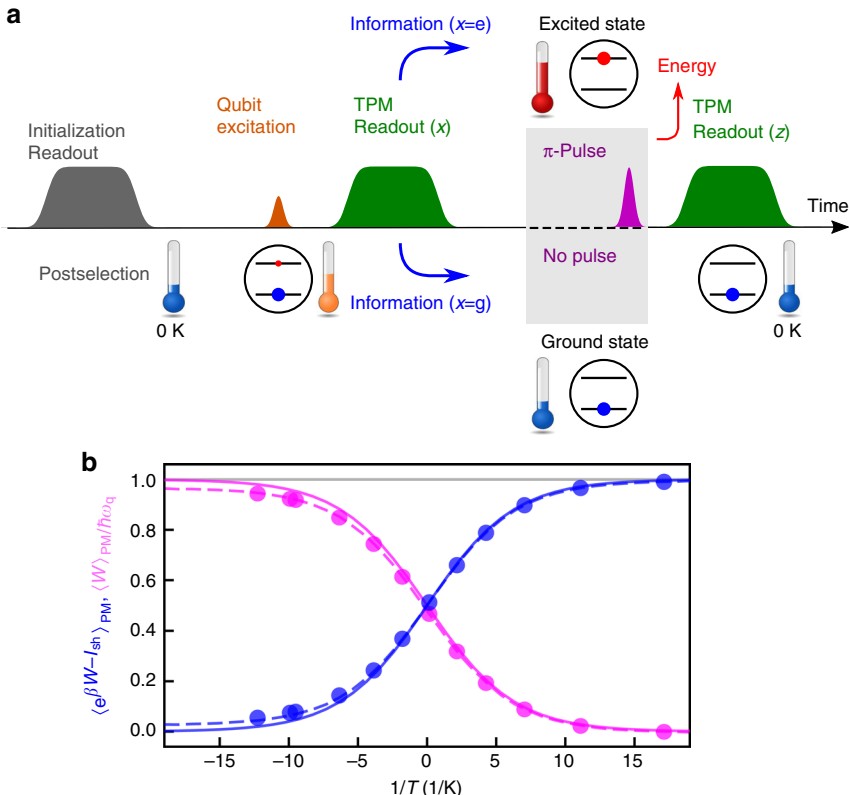

**Fig. 2** Generalized integral fluctuation theorem under feedback control. **a** Pulse sequence used in the experiment. The qubit is initialized with a projective measurement and postselection, followed by a resonant pulse excitation which prepares the qubit in a superposition state as an input. The two-point measurement protocol (TPM) involves two quantum non-demolition projective readout pulses. Depending on the outcome $x$ of the first readout ($x = g$ or $e$ corresponding to the ground or the excited state of the qubit), a $\pi$-pulse for the feedback control is or is not applied. The $\pi$-pulse flips the qubit state to the ground state and extracts energy. The second readout with outcome $z$ completes the protocol. See the Supplementary Note 2 for details. **b** Experimentally obtained statistical averages $\left\langle e^{\beta W - I_{sh}} \right\rangle_{PM}$ (blue circles) and $\langle W \rangle_{PM}/\hbar\omega_q$ (magenta circles) vs. the inverse initial qubit temperature $1/T$. The blue solid curve (gray solid line) is the theoretical value of the probability $1 - \lambda_{fb}$ in the presence (absence) of absolute irreversibility. The magenta solid curve is the expectation value of the normalized extracted work. The corresponding blue and magenta dashed curves are obtained by a master equation, which takes into account the qubit relaxation during the pulse sequence

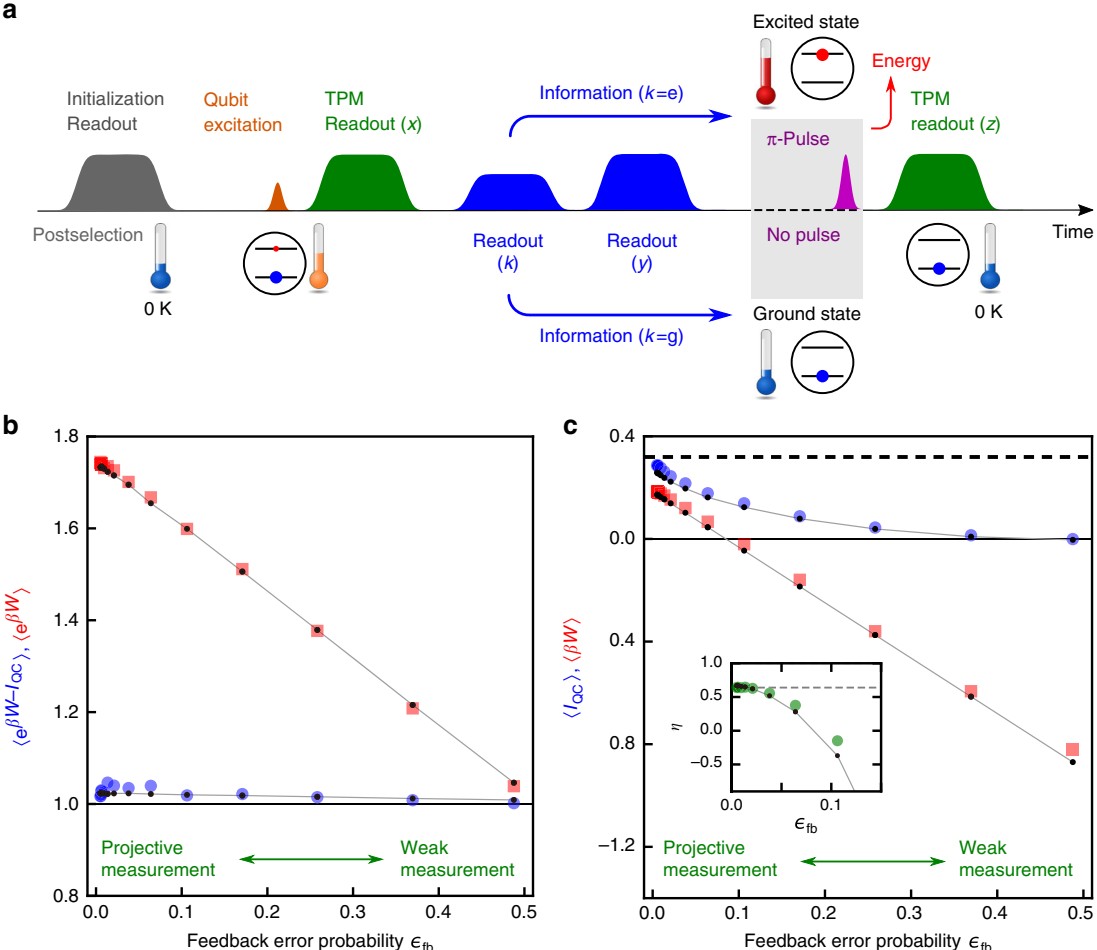

**Fig. 3** Effects of the feedback error on the fluctuation theorem and the second law of thermodynamics. **a** Pulse sequence. Two readout pulses are inserted between the two TPM pulses in Fig. 2a. The outcome $k(= g$ or $e)$ obtained by the readout with a variable pulse amplitude is used for the feedback control. The feedback error probability $\epsilon_{fb}$ is a function of the measurement strength. The subsequent readout with outcome $y$ projects the qubit state before the feedback control. See Supplementary Note 2 for details. **b** Experimentally determined $\langle e^{\beta W - I_{QC}} \rangle$ (blue circles) and $\langle e^{\beta W} \rangle$ (red squares) vs. the feedback error probability $\epsilon_{fb}$. **c** $\langle I_{QC} \rangle$ (blue circles) and $\langle \beta W \rangle$ (red squares) vs. $\epsilon_{fb}$. The black dashed line represents the Shannon entropy $\langle I_{Sh} \rangle_{PM}$ of the qubit initial state, which is prepared at the effective temperature $T = 0.14$ K with the excited state occupancy of 0.097. Line-connected black dots in **b**, **c** show the simulated results incorporating the effect of qubit relaxation (Supplementary Note 2). Inset in **c**: information-to-work conversion efficiency $\eta$ (green circles) and the simulated result (line-connected black dots). The gray dashed line indicates the value for the efficiency in the limit of the projective measurement due to the absolute irreversibility

with the theoretical value of $1 - \lambda_{fb}$ (blue solid curve), where $p(x, z)$ is the joint probability of observing a particular combination of the outcomes $x$ and $z$ (Supplementary Note 1). We also plot the normalized average work, $\langle W \rangle_{PM}/\hbar\omega_q$ (magenta circles), extracted in the protocol. Depending on the effective temperature of the qubit initial state, the probability of the absolutely irreversible events varies. The excellent agreement between theory and experiment confirms the generalized integral fluctuation theorem under feedback control. Furthermore, the relation in Eq. (1) is proven to hold for any initial effective temperature of the qubit, even at negative temperatures. The smaller the inverse temperature $\beta$ is, the larger the contribution of absolute irreversibility.

**Effect of imperfect projection.** Next, we investigate the effects of imperfect projection in the readout. With a weak readout pulse, the state of the qubit is not completely projected. It also gives less information gain for the feedback control. To evaluate the influence of the weak measurement, we add two more readout

pulses to the pulse sequence (Fig. 3a). The TPM again starts with a projective readout with outcome $x$, but now the feedback control is performed based on the subsequent variable-strength measurement with outcome $k(= g$ or $e)$. Then, to project the qubit state before the feedback control, we apply another strong measurement to obtain outcome $y(= g$ or $e)$. Using these measurement outcomes, we calculate the stochastic QC-mutual information $I_{QC}(x, k, y) = \ln p(y|k) - \ln p(x)$[9]. Here, QC indicates that the measured system is quantum and the measurement output is classical[2], and $p(y|k)$ is the probability of outcome $y$ being obtained conditioned on the preceding measurement outcome $k$. The first term in $I_{QC}$ quantifies the correction to $I_{Sh}$ because of the imperfect projection. If the measurement for the feedback control is a QND projective measurement and there is no relaxation of the qubit, $p(y|k)$ becomes unity and $I_{QC}$ reduces to $I_{Sh}$. On the other hand, for the measurement with imperfect projection, the absolute irreversibility disappears, because such measurement no longer gives restriction on forward events. Therefore, we obtain $\lambda_{fb} = 0$. In this case, the generalized integral

fluctuation theorem is reformulated as[9] (see also Supplementary Note 1)

$$\langle e^{\beta W - I_{QC}} \rangle = 1. \tag{2}$$

Figure 3b plots the statistical averages, $\langle e^{\beta W - I_{QC}} \rangle$ and $\langle e^{\beta W} \rangle$, evaluated from the measurement outcomes of the pulse sequence shown in Fig. 3a. For example, $\langle e^{\beta W - I_{QC}} \rangle$ is experimentally obtained as $\sum_{x,k,y,z} p(x,k,y,z) \, e^{\beta W(x,z) - I_{QC}(x,k,y)}$, where $p(x,k,y,z)$ is the joint probability of observing a combination of the outcomes. By changing the amplitude of the readout pulse, which measures $k$, it is possible to continuously vary the post-measurement state from the projected state to a weakly disturbed state. Accordingly, the feedback error probability $\epsilon_{fb}$ increases with decreasing the readout pulse amplitude. (See the Supplementary Note 2 for details.) We see that $\langle e^{\beta W - I_{QC}} \rangle$ (blue circles), which involves the information gain due to the measurement, is almost unity regardless of the feedback error probability. The small deviation from unity is understood as the effect of the qubit relaxation during the TPM as indicated by the simulated result (black dots and gray lines interpolating them)[25] (see Supplementary Note 2). In contrast, the value $\langle e^{\beta W} \rangle$ (red squares), which discards the information used in the feedback operation, clearly deviates from unity. For the weaker readout amplitude, however, the amount of information gain becomes less, and thus $\langle e^{\beta W} \rangle$ becomes closer to unity. This situation corresponds to the integral fluctuation theorem in the absence of feedback control.

Figure 3c depicts the statistical averages $\langle I_{QC} \rangle$ (blue circles) and $\langle \beta W \rangle$ (red squares) as functions of the feedback error probability $\epsilon_{fb}$. Here, $\langle I_{QC} \rangle$ is always larger than $\langle \beta W \rangle$ in accordance with the inequality, $\langle \beta W \rangle \leq \langle I_{QC} \rangle$, derived from the fluctuation theorem Eq. (2). The QC-mutual information $\langle I_{QC} \rangle$ decreases to zero with increasing $\epsilon_{fb}$. On the other hand, for $\epsilon_{fb} \to 0$, $\langle I_{QC} \rangle$ approaches $\langle I_{Sh} \rangle_{PM} = -\sum_x p(x) \ln p(x)$ (black dashed line). The remaining difference between $\langle I_{QC} \rangle$ and $\langle I_{Sh} \rangle$ is due to the qubit relaxation between the two readouts for $k$ and $y$.

**Conversion efficiency.** The conversion efficiency from the QC-mutual information $\langle I_{QC} \rangle$ to the work $\langle W \rangle$ is defined for $T > 0$ as[4, 5]

$$\eta = \frac{\langle W \rangle}{k_B T \langle I_{QC} \rangle}, \tag{3}$$

where we omit the contribution from the free-energy change by assuming $\Delta F = 0$. As shown in the inset of Fig. 3c, $\eta$ is 0.65 in the limit of $\epsilon_{fb} \to 0$ corresponding to the case with the projective measurement shown in Fig. 2.

The efficiency obtained with the projective measurement is to be compared with the following inequalities:

$$\langle W \rangle \leq \langle W \rangle_{PM} \leq k_B T \langle I_{Sh} \rangle_{PM} + k_B T \ln(1 - \lambda_{fb}). \tag{4}$$

The first inequality describes the fact that for a given protocol the extracted work with a proper projective measurement is superior to that obtained with an imperfect projection, which is demonstrated in Fig. 3c. On the other hand, the second inequality derived for $T > 0$ from the fluctuation theorem Eq. (1) represents the generalized second law of information thermodynamics (Supplementary Note 1). We find that the contribution from the absolute irreversibility sets the limit of the efficiency, given by $\eta = 1 - |\ln(1 - \lambda_{fb})|/\langle I_{Sh} \rangle_{PM}$, which is indicated by the dashed line in the inset of Fig. 3c. The experimental result demonstrates that our feedback scheme achieves the equality condition in Eq. (4) and is optimal (though not ideal) in this sense.

## Discussion

We have successfully implemented Maxwell's demon in a setup based on superconducting circuit quantum electrodynamics and verified the generalized integral fluctuation theorem in a single qubit. In the present work, the measurement outcome obtained by the demon was analyzed in terms of the Shannon and the QC-mutual information. On the other hand, the effect of the coherence can be investigated in a similar setup[26]. By implementing the memory of the demon with a qubit[27], or a quantum resonator as demonstrated recently[28], one can characterize the energy cost for the measurement[29] or study feedback schemes maintaining the coherence between the system and the memory to improve the energy efficiency of the feedback. Superconducting quantum circuits further allow us to extend the study of information thermodynamics to larger and more complex quantum systems. It will lead to an estimation of the lower bound of the thermodynamic cost for quantum information processing.

## Methods

**Sample.** The transmon qubit has the resonance frequency $\omega_q/2\pi = 6.6296$ GHz, the energy relaxation time $T_1 = 24$ μs, and the phase relaxation time $T_2^* = 16$ μs at the base temperature ~ 10 mK of a dilution refrigerator. The cavity has the resonance frequency $\omega_{cav}/2\pi = 10.6180$ GHz, largely detuned from the qubit, and the relaxation time $1/\kappa = 0.076$ μs. The coupling strength between the qubit and the resonator is estimated to be $g/2\pi = 0.14$ GHz.

**Pulse sequences.** The pulse sequences for the experiments in Figs. 2 and 3 take about 2.5 and 4 μs, respectively. Each readout pulse has the width of 500 ns. The qubit excitation pulse and the feedback control pulse are both 20 ns wide. See the Supplementary Note 2 for details. We take the statistics of the outcomes by repeating the pulse sequence about $8 \times 10^4$ times, with a repetition interval 300 μs, which is much longer than the qubit relaxation time.

**Data availability.** All the data used in this study are available from the corresponding author upon reasonable request.

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

## Acknowledgements

We acknowledge T. Sagawa for useful discussions and W.D. Oliver for providing the transmon qubit. This work was partly supported by JSPS KAKENHI (Grant Number 26220601), NICT, and JST ERATO (Grant Number JPMJER1601). Y. Murashita was supported by JSPS through the Program for leading Graduate School (MERIT) and JSPS Fellowship (Grant Number JP15J00410). K.F. acknowledges supports from the National Science Foundation of China (grants 11375012 and 11534002).

## Author contributions

Y. Masuyama, K.F., and Y. Murashita designed the experiments. Y. Masuyama conducted the experiments. S.K. and Y.T. assisted in setting up the measurement system. K.F., Y. Murashita, and M.U. provided theoretical supports. A.N., Y.T., and R.Y. participated in discussions on the analysis. Y. Masuyama and Y.N. wrote the manuscript with feedback from all authors. M.U. and Y.N. supervised the project.

## Additional information

**Competing interests:** The authors declare no competing interests.

