## [Peer Review File(PDF 316 kb) · Nature Communications]

Reviewers' comments:

Reviewer #1 (Remarks to the Author):

This paper describes the experimental implementation of a Maxwell's demon in a circuit-QED architecture. This experimental platform allows coherent control, quantum non-demolition projective measurement of a superconducting quantum bits and the ability to close feedback loops using fast electronics.

Combining all these elements, the authors verify the generalized integral fluctuation theorems. Moreover the concept of absolute irreversibility is nicely illustrated using a combination of weak and strong measurements. These two demonstrations are certainly very important towards the goal of bridging quantum information theory and thermodynamics.

The experimental results are really convincing and presented in a pedagogical way. Each set of data is compared to a simple analytical formula when possible or to numerical simulations based on a master equation approach. The data-theory agreement is always sound.

I might recommend publication if the authors can address the two following concerns appropriately:

One key point of the paper is the ability of the authors to prepare the qubit in a thermal state and thus infer a temperature assuming a Boltzmann distribution (as pointed out in part F of the supplementary information). This part, which is central in my opinion, deserves a better explanation (probably in the body of the paper). Moreover, as explained in part D of the supplementary information, the QNDness of the measurement is not perfect. Whether this experimental imperfection impacts or not the inferred temperature should be discussed as well. As pointed in the abstract of the paper, the circuitQED architecture provides very good and efficient coherent control on superconducting qubits. In addition the authors emphasize the fact that they use a quantum system and the associated quantum trajectories (" Here we verify the generalized integral theorem under feedback control by using a superconducting transmon qubit as a quantum system and taking statistics over repeated single-shot measurements on individual quantum trajectories."). My question is very simple: in the results presented in this paper, what is purely quantum and what can be explained by classical means?

I also have minor comments:

- When describing the two-point measurement protocol, the authors use the notation x and z for the first and second measurement respectively. However x and z are usually referred as the expectation values of the Pauli operators in the superconducting qubits community. For the sake of clarity I suggest to change this notation to A and B or 1 and 2 for example.

- According to the inset of figure 3.c the feedback procedure is optimal. I'm very surprised that feedback delay (which is usually the main limitation in circuitQED feedback schemes) doesn't prevent to reach such optimum. Could the authors comment on that?

Reviewer #2 (Remarks to the Author):

Masumaya et al provide an experimental demonstration of the generalized integral fluctuation theorem in the context of a Maxwell demon. The experiment is performed within a quantum setup involving a superconducting qubit undergoing QND measurements. The demonstrations rely on a detailed theoretical model provided in the Supplementary material. Such an implementation opens the way to the study of quantum features of information to energy conversion.

In the first part of the paper, the authors study the fluctuation theorem (FT) with perfect feedback, which involves the use of projective measurements. This ideal case leads to forbidden reversed

trajectories, i.e. absolute irreversibility and to potentially non optimal work extraction. The Fig.2 shows very clearly the behavior of the FT as a function of the effective initial temperature of the qubit.

In the second part of the paper, the authors focus on the influence of errors in the feedback process, that would be due to imperfect quantum measurement in the feedback loop. In this case absolute irreversibility vanishes as all reversed trajectories are now possible, even with low probability. The authors relate the amount of extracted work to the Shannon entropy and the absolute irreversibility, and show that work extraction is optimal in their case.

The topics of experimental quantum thermodynamics is very hot, the results of the manuscript are extremely solid, and the synergy between theory and experiment excellent. Indeed, it is the first experimental investigation of FT in the context of Maxwell's demon within a quantum setup. For these reasons, the paper should be published in Nature Comm. However, the manuscript is currently missing the pedagogy that is required for the broad audience of the journal. Essential pieces of information are currently spread in different places: Supplementary, captions, main text. Moreover, some sentences are contradictory. Here are some suggestions/questions to clarify the message:

- The concept of absolute irreversibility is essential to the paper. It would deserve to be properly introduced and explained in the beginning of the paper. The Formula (1) should be demonstrated in the textbook case of projective measurement, which also corresponds to the ideal implementation of the Maxwell's demon, in the beginning of the paper for the reader to grasp the basic and fundamental physics at play.
- The fact that in the ideal case of projective measurement there is a trade-off between the information acquired by the demon and the amount of work extracted (as clearly shown in eq. (4)) is very counter-intuitive and extremely interesting: In my opinion it should also appear much sooner in the paper, and be commented as a characteristics of an ideal Maxwell's demon.
- This trade-off could also be commented on Fig.2: the FT equals 1 (which potentially gives rise to maximal work extraction) if the temperature is low and positive, i.e. if the qubit is mostly in the state $|g\rangle$, which in principle does provide very little work. On the other hand, the FT equals 0 and work extraction is not optimal if the temperature is small and negative, i.e. if the qubit is initialized in the state $|e\rangle$ (which intuitively gives rise to maximal work): To grasp this physics better, could the authors also show the mean work extraction as a function of temperature to complete Fig.2?
- How comes that the energy of the qubit never appears as a relevant energy scale, especially in Eq.(4)?
- The caption of Fig.1 is not clear: It is stated that "an unoptimized feedback operation prevents it and introduces absolute irreversibility". On the contrary, in the main text it is stated "The absolute irreversibility is caused by the projective measurements that restricts possible forward events". The authors should precize what they mean by optimized feedback operation. Fig.3 shows very clearly that maximal efficiencies correspond to projective measurements, i.e. error free feedback - even if it leads to absolute irreversibility. Moreover, why should the final state be identical to the initial state to achieve the ultimate bound of work extraction? These points should be explained in the main text.
- The definition of the efficiency (3) used by the authors is not standard. In general in the litterature, the efficiency is defined as the net work extracted, divided by the resource. The net work extracted should be the work provided by the qubit, minus the work used to erase the memory of the demon. The resource is the heat absorbed by the qubit in the first step. To be consistent with other implementations or proposals, the authors should change their definition, or argue why they use this one.
- To study the influence of errors on the FT, the authors have chosen an initial temperature ($T=0.14K$) of the qubit giving rise to $\langle \exp(-\beta W - I_{QC}) \rangle = 1$, even in the projective measurement case. How does the plot change if the authors start from a larger temperature? From negative temperatures? Can we see the FT continuously evolving from $1-\lambda$ to 1 when the error rate increases?

- The experimental setup should be briefly presented in the main text, and avoid jargon (e.g., "The pi-pulse")
- In the authors' opinion, are there specifically quantum effects due to coherences in the presented experiment, or does this experiment rather provide a first step towards studying this new physics?
- In particular, what is the interest of initializing the qubit in a quantum superposition? As the demon reads in the energy basis of the qubit, the superposition is immediately collapsed by the mere action of the demon.

Reviewer #3 (Remarks to the Author):

I find that the main results of the paper are a little abstract or disconnected from the actual experiment and/or the actual measurement results. As an example in reading both the paper and the supplemental section, it is not clear in Fig. 2b., how the effective temperature of the system is being changed or how the expectation values are extracted on the vertical axis. Perhaps this is a result of myself not working in the field of information extraction (in particular for the extracted expectation values), but I would encourage the authors to ask themselves how exactly did they measure and extract the different data in the figures and did they explain it in the simplest, while being technically accurate, possible way.

Here are some specific recommendations:

1. I recommend making it clear in the main text how the inverse of the effective temperature is defined and changed (e.g. negative Kelvin values). I think this is alluded to in the supplemental section.
2. A few comments on figures 3b and 3c.
 - a. At first I didn't understand the difference between the fainter and larger data points (measured values) and the smaller data points (calculated values). I recommend distinguishing them better or be more descriptive in the figure caption.
 - b. Are there many data points for small feedback error probabilities? I don't think so but if so perhaps there is a better way to graph the data (semi-log plot?).
 - c. Is there another way you can denote the strength of feedback error probability, which is primarily discussed in the supplemental section, in terms of a direct experimental parameter?

Reply to Reviewer 1:

First of all, we thank the Reviewer for his/her careful examination of our paper.

1. *This paper describes the experimental implementation of a Maxwell's demon in a circuit-QED architecture. This experimental platform allows coherent control, quantum non-demolition projective measurement of a superconducting quantum bits and the ability to close feedback loops using fast electronics. Combining all these elements, the authors verify the generalized integral fluctuation theorems. Moreover the concept of absolute irreversibility is nicely illustrated using a combination of weak and strong measurements. These two demonstrations are certainly very important towards the goal of bridging quantum information theory and thermodynamics. The experimental results are really convincing and presented in a pedagogical way. Each set of data is compared to a simple analytical formula when possible or to numerical simulations based on a master equation approach. The data-theory agreement is always sound. I might recommend publication if the authors can address the two following concerns appropriately:*

We thank the Reviewer for the concise summary and positive evaluation of our manuscript.

2. *One key point of the paper is the ability of the authors to prepare the qubit in a thermal state and thus infer a temperature assuming a Boltzmann distribution (as pointed out in part F of the supplementary information). This part, which is central in my opinion, deserves a better explanation (probably in the body of the paper). Moreover, as explained in part D of the supplementary information, the QNDness of the measurement is not perfect. Whether this experimental imperfection impacts or not the inferred temperature should be discussed as well.*

We appreciate the suggestion by the Reviewer. As the Reviewer implied, we did not prepare a Boltzmann distribution before the two-point measurement protocol (TPM). Instead, we prepared a coherent superposition of $|g\rangle$ and $|e\rangle$, say, $\alpha_g|g\rangle + \alpha_e|e\rangle$ ($|\alpha_g|^2 + |\alpha_e|^2 = 1$) by the initial qubit excitation pulse. The first projective measurement in the TPM statistically turned the two-level system into a thermal state with the effective temperature $T = (\hbar\omega_q/k_B)/\ln[|\alpha_g|^2/|\alpha_e|^2]$. Thus, T could be adjusted by the

amplitude of the initial qubit excitation pulse and could be even negative. Though we could have had an additional projective readout pulse before the two-point measurement protocol (TPM) to make the preparation procedure more explicit, we had omitted it for simplicity in the experimental pulse sequence.

The impact of the imperfection of the qubit readout to the evaluation of the effective temperature is negligibly small. The imperfection of QNDness presented in Sec. IID in the Supplementary Information, 0.4% for the ground state population and 3.4% for the excited state, gives an upper limit of the error in the evaluation of the temperatures in Fig. 2b. The amount of the error is estimated to be at most a few percent and smaller than the size of the data points.

According to the suggestion by the Reviewer, we added an explanation about the state preparation at the end of page 2 of the manuscript.

3. *As pointed in the abstract of the paper, the circuit QED architecture provides very good and efficient coherent control on superconducting qubits. In addition the authors emphasize the fact that they use a quantum system and the associated quantum trajectories (“Here we verify the generalized integral theorem under feedback control by using a superconducting transmon qubit as a quantum system and taking statistics over repeated single-shot measurements on individual quantum trajectories.”). My question is very simple: in the results presented in this paper, what is purely quantum and what can be explained by classical means?*

As mentioned above in the reply to item 2, the TPM protocols prepare the states without qubit coherence by the first projective measurement (with the outcome x) in the energy eigenbasis. We also chose protocols which did not generate coherence in the final state, either. Thus, the evolution of the system and the expectation of the measurement outcomes did not explicitly show any dynamics unique to quantum theory and were describable with the classical probability theory. By limiting ourselves to such simple protocols, we were able to utilize existing theories to analyze the experiments, which clarified the connection of the quantum measurement and feedback control with thermodynamics.

Nevertheless, the system is quantum in the sense that it has discrete energy levels

and is subject to the QND measurement relying on the quantum properties of the system. Therefore, it is clearly distinguished from the preceding works on classical systems [Refs.4-6 in the manuscript]. Moreover, the present work can be extended to the situations with coherence in the initial state and/or generated coherence in the final state. We may also apply a similar scheme to a multiple qubit system with entanglement. Considering the recent theoretical and experimental progresses in quantum information thermodynamics, we expect future development of the approach.

4. *I also have minor comments: When describing the two-point measurement protocol, the authors use the notation x and z for the first and second measurement respectively. However x and z are usually referred as the expectation values of the Pauli operators in the superconducting qubits community. For the sake of clarity I suggest to change this notation to A and B or 1 and 2 for example.*

We appreciate the suggestion by the Reviewer. However, let us keep the notations for the moment, partly to avoid confusion in the second round of the review. As far as we are aware of, people in the community use more often X , Y , and Z , or σ_x , σ_y and σ_z to represent Pauli operators, and $\langle X \rangle$ (sometimes X itself) or $\langle \sigma_x \rangle$, and so on, for their expectation values. If the Reviewer(s) and the Editor would consider it necessary, we would be happy to replace later x , z , k and y with a , b , k and l , respectively, for example.

5. According to the inset of figure 3.c the feedback procedure is optimal. I'm very surprised that feedback delay (which is usually the main limitation in circuitQED feedback schemes) doesn't prevent to reach such optimum. Could the authors comment on that?

The minimum response time of the feedback loop was about 200 ns, and we set an enough delay time between the relevant measurement and control in the protocol. The delay time of about 1 μ s was to be compared with the relaxation time of the qubit of about 24 μ s. Thus, the effect was minimal, as seen in the calculation based on the master equation shown in the inset of Fig. 3c.

Reply to Reviewer 2:

We thank the Reviewer for his/her careful examination of our paper and for the valuable comments.

1. *Masuyama et al. provide an experimental demonstration of the generalized integral fluctuation theorem in the context of a Maxwell demon. The experiment is performed within a quantum setup involving a superconducting qubit undergoing QND measurements. The demonstrations rely on a detailed theoretical model provided in the Supplementary material. Such an implementation opens the way to the study of quantum features of information to energy conversion.*

In the first part of the paper, the authors study the fluctuation theorem (FT) with perfect feedback, which involves the use of projective measurements. This ideal case leads to forbidden reversed trajectories, i.e. absolute irreversibility and to potentially non optimal work extraction. The Fig. 2 shows very clearly the behavior of the FT as a function of the effective initial temperature of the qubit.

In the second part of the paper, the authors focus on the influence of errors in the feedback process, that would be due to imperfect quantum measurement in the feedback loop. In this case absolute irreversibility vanishes as all reversed trajectories are now possible, even with low probability. The authors relate the amount of extracted work to the Shannon entropy and the absolute irreversibility, and show that work extraction is optimal in their case.

The topics of experimental quantum thermodynamics is very hot, the results of the manuscript are extremely solid, and the synergy between theory and experiment excellent. Indeed, it is the first experimental investigation of FT in the context of Maxwell's demon within a quantum setup. For these reasons, the paper should be published in Nature Comm.

We thank the Reviewer for the positive assessment of our manuscript.

2. *However, the manuscript is currently missing the pedagogy that is required for the broad audience of the journal. Essential pieces of information are currently spread in*

different places: Supplementary, captions, main text. Moreover, some sentences are contradictory.

Following the comments by the Reviewers, we modified the manuscript to make it as clear as possible. We believe that now the manuscript is much improved in its readability thanks to the suggestions by the Reviewers. See the revised manuscript and the answers to the following questions and comments.

3. *Here are some suggestions/questions to clarify the message: The concept of absolute irreversibility is essential to the paper. It would deserve to be properly introduced and explained in the beginning of the paper. The Formula (1) should be demonstrated in the textbook case of projective measurement, which also corresponds to the ideal implementation of the Maxwell's demon, in the beginning of the paper for the reader to grasp the basic and fundamental physics at play.*

We thank the Reviewer for the suggestion. We added a paragraph to explain the absolute irreversibility in page 2 before describing the experimental details. We also explained explicitly a “textbook” demonstration of the Maxwell’s demon by using Fig. 1a in the same paragraph.

4. *The fact that in the ideal case of projective measurement there is a trade-off between the information acquired by the demon and the amount of work extracted (as clearly shown in eq. (4)) is very counter-intuitive and extremely interesting: In my opinion it should also appear much sooner in the paper, and be commented as a characteristics of an ideal Maxwell's demon.*

We would not call it a trade-off on the basis of the following reasons.

We should keep in mind that the equalities derived from the fluctuation theorems cannot always be achieved. In our case, under the feedback protocol described in the main text, the equality of $\langle W \rangle_{\text{PM}} \leq k_{\text{B}}T \langle I_{\text{Sh}} \rangle_{\text{PM}} + k_{\text{B}}T \ln(1 - \lambda_{\text{fb}})$ [Eq.(4) in the previous version, with the new notation of the bracket where $\langle \cdot \rangle_{\text{PM}}$ represents the statistical average taken with a feedback protocol using projective measurement.] can be achieved (except for negligible effects from the unwanted qubit relaxation), while the equality

of $\langle W \rangle \leq k_B T \langle I_{QC} \rangle$ cannot. Therefore, the latter inequality fails to give the maximum work extraction. To clarify this, we modified Eq.(4) and included another inequality on the left. Projective measurements onto a proper basis could always be ideal for extracting information $\langle I_{Sh} \rangle_{PM}$ from the quantum system. On the other hand, the term $k_B T \ln(1 - \lambda_{fb}) (\leq 0)$ in Eq.(4) is due to the feedback operation not ideally customized to the measurement for maximal work extraction. The absolute irreversibility is the concept to describe the non-ideal property of the feedback operation and gives a tighter bound, Eq.(4), to the maximal work extraction than $\langle W \rangle_{PM} \leq k_B T \langle I_{Sh} \rangle_{PM}$. We mentioned the important role of the feedback operation in the absolute irreversibility in the paragraphs newly added on page 2 in the main text.

For the case with imperfect projection, the absolute irreversibility vanishes and the generalized second law of thermodynamics becomes $\langle W \rangle \leq k_B T \langle I_{QC} \rangle$. However, this does not mean that the non-ideal property of the feedback operation disappears, but just indicates that the inequality bound becomes less tight.

5. *This trade-off could also be commented on Fig. 2: the FT equals 1 (which potentially gives rise to maximal work extraction) if the temperature is low and positive, i.e., if the qubit is mostly in the state $|g\rangle$, which in principle does provide very little work. On the other hand, the FT equals 0 and work extraction is not optimal if the temperature is small and negative, i.e. if the qubit is initialized in the state $|e\rangle$ (which intuitively gives rise to maximal work): To grasp this physics better, could the authors also show the mean work extraction as a function of temperature to complete Fig. 2?*

Let us emphasize again that it is not a trade-off. The temperature dependence observed in Fig. 2 indicates how close the feedback operation in the experiment is to the optimal one for the particular initial distribution of the qubit states at each temperature.

If the temperature is low and positive ($1/T \rightarrow +\infty$), i.e., if the qubit is mostly in the state $|g\rangle$, the average work extracted is small ($\langle W \rangle_{PM} \rightarrow 0$). However, the feedback control bringing the final state of the qubit back to $|g\rangle$ is efficient in the sense it achieves the limit $\langle W \rangle_{PM} = k_B T \langle I_{Sh} \rangle_{PM}$. FT equals 1 and the absolute irreversibility vanishes, i.e., $\lambda_{fb} = 0$, as pointed out by the Reviewer.

On the other hand, if the temperature is small and negative ($1/T \rightarrow -\infty$), the feedback control is least optimal for achieving $\hat{\rho}_{\text{fin}} = \hat{\rho}_{\text{ini}}$ (See also the reply to item 7 below). That is why λ_{fb} is largest and FT approaches zero in the limit.

Indeed, the second inequality in Eq.(4) holds only for $T > 0$. For $T < 0$, the corresponding formula reads

$$\langle W \rangle_{\text{PM}} \geq k_{\text{B}}T \langle I_{\text{Sh}} \rangle_{\text{PM}} + k_{\text{B}}T \ln(1 - \lambda_{\text{fb}}),$$

which gives a limitation only to the work injected to the system, but not to the work extracted from the system.

We added a phrase “for $T > 0$ ” to restrict the applicable temperature range of the inequality [Eq.(4)] in the sentence after the formula.

As the Reviewer mentioned, $\langle W \rangle_{\text{PM}}$ is a monotonically decreasing function of $1/T$. Theoretically, the dependence is given as $\langle W \rangle_{\text{PM}} = \hbar\omega_{\text{q}} e^{-\beta\hbar\omega_{\text{q}}} / (1 + e^{-\beta\hbar\omega_{\text{q}}})$ in the absence of qubit relaxation, which is nothing but the excited state probability of the qubit at the beginning of the TPM protocol. We added the plot of experimentally obtained $\langle W \rangle_{\text{PM}} / \hbar\omega_{\text{q}}$ and the theoretical curve in Fig. 2(b).

6. *How comes that the energy of the qubit never appears as a relevant energy scale, especially in Eq.(4)?*

The qubit energy $\hbar\omega_{\text{q}}$ is hidden in the expression. The value of W in each sequence takes 0 or $\pm\hbar\omega_{\text{q}}$. The expectation value $\langle W \rangle$, however, is compared with the thermal energy $k_{\text{B}}T$ multiplied with the information content without an explicit appearance of the qubit energy. Note that the temperature here is also defined in the unit of the qubit energy, i.e., $T = (\hbar\omega_{\text{q}}/k_{\text{B}}) / \ln[|\alpha_{\text{g}}|^2/|\alpha_{\text{e}}|^2]$. [See item 2 of the reply to Reviewer 1.]

7. *The caption of Fig. 1 is not clear: It is stated that “an unoptimized feedback operation prevents it and introduces absolute irreversibility”. On the contrary, in the main text it is stated “The absolute irreversibility is caused by the projective measurements that restricts possible forward events”. The authors should precise what they mean by optimized feedback operation. Fig. 3 shows very clearly that maximal efficiencies correspond to projective measurements, i.e. error free feedback - even if it leads to absolute*

irreversibility. Moreover, why should the final state be identical to the initial state to achieve the ultimate bound of work extraction? These points should be explained in the main text.

We thank the Reviewer for the questions. The sentence in the caption of Fig. 1 was correct. The other in the main text was incomplete. There are two independent factors in the problem: the perfectness of the projective measurement and that of the feedback operation. They should be considered separately.

Even though in the present setup the projective measurements result in the maximum efficiencies, it is not optimal in the sense that the protocol is still limited by the upper limit due to the absolute irreversibility (the gray zone in the inset of Fig. 3). This limitation comes not directly from the projective measurement, but from the non-ideal feedback operation in the protocol. We corrected the relevant sentence on page 2 of the manuscript.

There is a way to mitigate the limitation with an optimized feedback operation, though it is not straightforward to implement it in our setup with the fixed qubit excitation energy. The optimal scheme would require quasi-adiabatic control of the energy levels of the qubit.

To achieve the ultimate bound of work extraction, i.e., $\langle W \rangle_{\text{PM}} \leq k_{\text{B}}T \langle I_{\text{Sh}} \rangle_{\text{PM}}$, the final state should be in a canonical distribution at the same temperature as in the initial state. This is the necessary condition for the second law of information thermodynamics to achieve its equality as described in J. M. Holowitz and J. M. Parrando, *Europhys. Lett.* **95**, 10005 (2011) for classical systems and K. Funo *et al.* *New. J. Phys.* **17**, 075005 (2015) [Ref.8 in the manuscript] for quantum systems. Here, we additionally assumed that the final system Hamiltonian was not altered from the initial one. Thus, the final state should be identical to the initial state. We added a sentence to explain this in page 2 of the main text.

8. *The definition of the efficiency (3) used by the authors is not standard. In general in the literature, the efficiency is defined as the net work extracted, divided by the resource. The net work extracted should be the work provided by the qubit, minus the work used to erase the memory of the demon. The resource is the heat absorbed by the*

qubit in the first step. To be consistent with other implementations or proposals, the authors should change their definition, or argue why they use this one.

We agree that the efficiency suggested by the Reviewer is the standard one when we are interested in the entropy balance of the entire system including the demon. However, in the present manuscript, we are concerned with the conversion efficiency from the information obtained in the measurement to the extracted work. The definition of Eq.(3) is justified in this context and has been used in the literature on classical systems, for example, in S. Toyabe *et al.*, Nature Phys. **6**, 988 (2010) and J. V. Koski *et al.*, Phys. Rev. Lett. **113**, 030601 (2014). We cited the papers just before the definition of Eq.(3).

9. *To study the influence of errors on the FT, the authors have chosen an initial temperature ($T = 0.14$ K) of the qubit giving rise to $\langle \exp(-\beta W - I_{\text{QC}}) \rangle = 1$, even in the projective measurement case. How does the plot change if the authors start from a larger temperature? From negative temperatures? Can we see the FT continuously evolving from $1 - \lambda$ to 1 when the error rate increases?*

As Eq.(2), i.e., $\langle e^{\beta W - I_{\text{QC}}} \rangle = 1$, suggests, we do not expect any significant change in the result even at higher or negative effective temperature of the qubit. The slight deviation of FT [the left-hand side of Eq.(2)] from unity is understood as a result of the finite qubit relaxation rate, as the calculation based on the master equation indicates (Fig. 3b). As long as the qubit relaxation probability is small during the TPM protocol, we do not expect any significant temperature dependence other than the small contribution from the qubit relaxation.

We would not see continuous evolution of FT from $1 - \lambda_{\text{fb}}$ to unity even if the error rate increases. Equation (2) holds as long as the error rate ϵ_{fb} is finite. Only when ϵ_{fb} is zero, λ_{fb} becomes non-zero. In this case, as $1/T$ increases from $-\infty$ to ∞ , $\langle e^{\beta W - I_{\text{QC}}} \rangle (= \langle e^{\beta W - I_{\text{Sh}}} \rangle_{\text{PM}})$ evolves from 0 to 1, as shown in Fig. 2.

The apparent jump of λ_{fb} in the limit of $\epsilon_{\text{fb}} \rightarrow 0$ is a little counter-intuitive. The disappearance of λ_{fb} is formally absorbed in the difference between $I_{\text{Sh}} = -\ln p(x)$ and $I_{\text{QC}} = \ln p(y|k) - \ln p(x)$ in the case with finite feedback error probability ϵ_{fb} . When $y \neq k$, the first term $\ln p(y|k)$ in I_{QC} gives negative logarithmic divergence

in the limit of $\epsilon_{\text{fb}} \rightarrow 0$ and thus contributes to the ensemble in a singular manner. Actually, its contribution can formally be estimated to be nonvanishing as $p(y, k)e^{-I_{\text{QC}}} \simeq p(y|k)e^{-\ln p(y|k)} \rightarrow \text{const.}$ (for $\epsilon_{\text{fb}} \rightarrow 0$) [cf. Supplementary Material, Ref. 25].

A natural question arises concerning the imperfection in the projective measurement in Fig. 2. Indeed, in the protocol used in Fig. 2, the imperfection affects the evaluation of $\langle e^{\beta W - I_{\text{Sh}}} \rangle$ in the same manner as the qubit relaxation does. However, as long as the information content is evaluated by I_{Sh} (rather than I_{QC}), errors caused by the measurement imperfection do not alter the result significantly. Recall that the singular contribution that removes the absolute irreversibility originates from the term $\ln p(y|k)$ in I_{QC} . As most of the measurement imperfection is attributed to the qubit relaxation during the readout pulse and is already modeled in the master equation, the effect is rather small as indicated by the difference between the red solid curve and the green dashed curve in Fig. 2(b). This protocol corresponds to discarding the outcome y in the protocol in Fig. 3. The imperfect projection in the measurement of k and the feedback control based on the outcome can be translated to the relaxation of the qubit during the measurement of x .

10. *The experimental setup should be briefly presented in the main text, and avoid jargon (e.g., “The pi-pulse”)*

We modified the manuscript according to the suggestion by the Reviewer. A paragraph explaining the experimental setup was added in page 2 of the manuscript. We also avoided using jargons without definition, while we kept the word “ π -pulse” after mentioning qubit control with a resonant microwave pulse which induces Rabi rotation. We also mentioned explicitly that it flips the qubit state.

11. *In the authors’ opinion, are there specifically quantum effects due to coherences in the presented experiment, or does this experiment rather provide a first step towards studying this new physics?*

The answers are no and yes, respectively. Please refer to the answer to the related

question by Reviewer 1 (item 3).

12. *In particular, what is the interest of initializing the qubit in a quantum superposition? As the demon reads in the energy basis of the qubit, the superposition is immediately collapsed by the mere action of the demon.*

As the Reviewer pointed out, the superposition did not play any direct role in the protocol. It was used to prepare the mixed states with different effective temperatures by the projective measurement. Please refer to the answer (item 2) to the comment by Reviewer 1.

Reply to Reviewer 3:

1. *I find that the main results of the paper are a little abstract or disconnected from the actual experiment and/or the actual measurement results. As an example in reading both the paper and the supplemental section, it is not clear in Fig. 2b., how the effective temperature of the system is being changed or how the expectation values are extracted on the vertical axis. Perhaps this is a result of myself not working in the field of information extraction (in particular for the extracted expectation values), but I would encourage the authors to ask themselves how exactly did they measure and extract the different data in the figures and did they explain it in the simplest, while being technically accurate, possible way.*

We thank the Reviewer for careful reading of our manuscript. In the revised manuscript, we followed his/her suggestions to improve the clarity of the paper. For example, we added in the relevant paragraphs explicit formulae explaining how we evaluated the ensemble averages appearing in Figs. 2 and 3.

2. *Here are some specific recommendations: I recommend making it clear in the main text how the inverse of the effective temperature is defined and changed (e.g. negative Kelvin values). I think this is alluded to in the supplemental section.*

Following the suggestions by the Reviewer and other Reviewers, we added an explanation in the end of page 2 of the manuscript. See also item 2 in the reply to Reviewer 1.

3. *A few comments on figures 3b and 3c. a. At first I didn't understand the difference between the fainter and larger data points (measured values) and the smaller data points (calculated values). I recommend distinguishing them better or be more descriptive in the figure caption.*

We thank the Reviewer for pointing out this issue. We changed the color of the symbols and lines in Figs. 3b and 3c for better visibility.

4. *b. Are there many data points for small feedback error probabilities? I don't think so but if so perhaps there is a better way to graph the data (semi-log plot?).*

Yes, there are more data points in the small error limit than in the opposite limit in Figs. 3b and 3c. The distribution of the data points can be found in Fig. S6 of the Supplementary Information. However, those nearly degenerate data points in the small error limit do not have any independent meaning. Thus, we consider that it is not worth clearly resolving them in the plot. A semi-log plot would also hinder the linear dependences of $\langle e^{\beta W} \rangle$ and $\langle \beta W \rangle$ on ϵ_{fb} .

5. *c. Is there another way you can denote the strength of feedback error probability, which is primarily discussed in the supplemental section, in terms of a direct experimental parameter?*

We do not have any alternative idea, in terms of the simplicity and fairness, other than the current definition. We believe that our operational definition of the error probability is well justified.

List of changes in the figures in the resubmitted manuscript:

We modified the style of the main text to fit into the journal style. Other modifications of the text are highlighted in red both in the main text and the Supplementary Information.

List of changes in the figures in the resubmitted manuscript:

1. Figure 1:

In Fig. 1(a), labels to the arrows were added to clarify the corresponding processes. The colors of arrows were also changed for clarity.

Accordingly the figure caption was slightly modified.

Panel c is merged to panel b. The arrow indicating the measurement process was modified to a curved one to represent the microwave reflection measurement.

2. Figure 2:

According to the change made in the text, the label of the vertical axis of Fig. 2(b) was modified.

Following Reviewer's suggestion, we added the plot of experimentally obtained $\langle W \rangle_{\text{PM}}/\hbar\omega_q$ and the corresponding theoretical curves in Fig. 2(b).

The labels in Fig. 2(a) were modified slightly for clarity. The illustration of fire was removed.

3. Figure 3:

To improve the visibility, the color of the dots and lines for the simulation results in (b) and (c) were changed to black and gray, respectively.

The labels in Fig. 3(a) were modified slightly for clarity. The illustration of fire was removed.

The gray zone in the inset of Fig. 3(c) is changed to the gray dashed line indicating the maximal achievable efficiency in the limit of the projective measurement.

4. Figure S1:

The colors of arrows were also changed for clarity.

5. Figure S6:

The legend in (b) was corrected to represent conditional error probabilities.

REVIEWERS' COMMENTS:

Reviewer #1 (Remarks to the Author):

The authors addressed my comments and concerns in a convincing way. I thus recommend publication of the paper in Nature Communications.

Reviewer #2 (Remarks to the Author):

I am fully satisfied by the revisions made by the authors. This is a beautiful manuscript and I am sure it will attract a broad audience of readers. I strongly support its publication in Nature Communications.